# Label-Free Aptasensor for Detection of Fipronil Based on Black Phosphorus Nanosheets

**DOI:** 10.3390/bios12100775

**Published:** 2022-09-20

**Authors:** Hao Huang, Chuanxiang Zhang, Jie Zhou, Dan Wei, Tingting Ma, Wenfei Guo, Xueying Liu, Song Li, Yan Deng

**Affiliations:** 1Hunan Key Laboratory of Biomedical Nanomaterials and Devices, Hunan University of Technology, Zhuzhou 412007, China; 2College of Packing and Materials Engineering, Hunan University of Technology, Zhuzhou 412007, China

**Keywords:** fipronil, black phosphorus nanosheets, aptamers, Au nanoparticles, electrochemical biosensors

## Abstract

A label-free fipronil aptasensor was built based on Polylysine-black phosphorus nanosheets composition (PLL-BPNSs) and Au nanoparticles (AuNPs). A PLL-BP modified glassy carbon electrode (GCE) was fabricated by combining BP NSs and PLL, which included a considerable quantity of -NH_2_. Au nanoparticles (AuNPs) were placed onto the GCE, and PLL-BPNSs bonded to Au NPs firmly by assembling. The thiolated primers were then added and fixed using an S-Au bond, and competitive binding of the fipronil aptamer was utilized for fipronil quantitative assessment. The sensor’s performance was evaluated using differential pulse voltammetry (DPV) method. The linear equation is ΔI (μA) = 13.04 logC + 22.35, while linear correlation coefficient R^2^ is 0.998, and detection limit is 74 pg/mL (0.17 nM) when the concentration of fipronil is 0.1 ng/mL–10 μg/mL. This aptasensor can apply to quantitative detection of fipronil.

## 1. Introduction

Pesticides have made significant contributions to pest control and agricultural output [1]. Fipronil (FP) is a very effective phenylpyrazole insecticide and is now widely used in agriculture [2]. FP is a process-targeted pesticide that has remarkable insect selectivity. However, inadvertent contact, incorrect use of FP, or widespread use of FP may pollute water and soil, causing a range of harmful consequences in animals and people, including neurotoxicity, hepatotoxicity, nephrotoxicity, and reproductive and cytotoxic effects in vertebrates and invertebrates [3,4,5,6]. Headaches, dizziness, sweating, and other symptoms have been described because of fipronil poisoning in humans [7]. Currently, the most common techniques for detecting fipronil are gas chromatography-mass spectrometry (GC-MS), liquid chromatography-mass spectrometry (LC-MS), and gas chromatography (GC) [8,9,10]. However, these traditional methods have challenges, such as complex sample pretreatment, high cost, and dependence on trained staff. As a result, it is critical to design a technique for detecting fipronil residues that is both quick and easy. Biosensors, as an alternate tool for pesticide detection, have received a lot of attention in recent decades. Electrochemical biosensors, as a cost-effective and portable analytical approach, overcome the limitations of traditional methods by providing high sensitivity, selectivity, and response time, as well as demonstrating significant potential in field tests. Black phosphorus nanosheets (BPNSs) are a new two-dimensional material with several distinguishing characteristics, including great biocompatibility, outstanding anisotropy and conductivity, high carrier mobility, and a programmable energy band gap [11,12,13,14]. BPNSs are often used for sensor preparation and application due to their great performance, and can be prepared by mechanical exfoliation, liquid exfoliation, chemical synthesis, and electrochemical exfoliation [15,16,17,18]. However, black phosphorus is unstable and easily degraded in air and water, and this degradation will be intensified under visible light irradiation, which also limits the application of black phosphorus [19]. The modification and functionalization of BPNSs improved the sensor’s stability, sensitivity, selectivity, and biocompatibility. Polymer poly-L-lysine (PLL) is a cationic polymer with good biocompatibility. The non-covalent electrostatic interaction between BPNSs and PLL not only retains the original wrinkled honeycomb structure of BP, but also enhances the stability and dispersion of BP in aqueous solution, which provides the possibility for preparation of stable biosensors [20,21].

Aptamers (Apt) are a class of oligonucleotide fragments screened from the oligonucleotide library that can identify targets [22,23], which show a very high affinity for their targets, comparable to those of some monoclonal antibodies, sometimes even better [24]. The fitness has a number of advantages over antibodies, including ease of manufacture, cheap production costs, minimal batch differences, reversible folding properties, and low immunogenicity [25]. However, the true value of the aptamer lies in its simplicity; these molecules may be built to function as sensors, actuators, and other devices, which are typically at the heart of new innovations [26,27,28,29,30].

Au nanoparticles (AuNPs) are widely believed to improve the detection speed and stability of biosensors [31]. AuNPs can increase DNA immobilization and signal amplification on electrode surface, enhancing the modified surface’s hybridization capacity for sensitivity detection [32,33,34]. In summary, the goal of this study was to develop an aptamer sensor based on BPNSs with high sensitivity and specificity for detecting the presence of fipronil in agricultural residues, using PLL-BP and AuNPs modified glassy carbon electrodes to build a sensitive sensing platform. The Fipronil aptamer was efficiently fixed using a thiolated primer modified sensing platform, and then the fipronil aptamer was modified to precisely recognize fipronil in the sample. Differential pulse voltammetry (DPV) was used to determine the amount of fipronil in the sample. The sensor offers outstanding repeatability, stability, and specificity, as well as a wide range of applications.

## 2. Materials and Methods

### 2.1. Materials and Reagents

Mercaptoethanol (MCH) was purchased from Tokyo Chemical Industry Co., Ltd. (Shanghai, China). Fipronil (Fp), aldicarb sulfoxide (As), fenitrothion (Ft), carbofuran (Cf), and aldicarb (Ac) were obtained from Tanmo Quality Inspection Technology Co., Ltd. Tris-(2-carboxyethyl)-phosphine hydrochloride (TCEP), Tween 20, and polylysine (PLL) were obtained from Bioengineering Shanghai Co., Ltd. Potassium hexacyanoferrate(III) (K_3_Fe(CN)_6_), and potassium chloride (KCl) were obtained from Sinopharm Group Chemical Reagent Co., Ltd. (Shanghai, China). All the reagents were of analytical grade. The required aptamer sequence [TGTACCGTCTGAGCGATTCGTACAGTTTCTGGAGGACTGGGCGGGGTGACGGTTATGAGCAGTCAGTGTTAAGGAGTGC] and thiolated primers [GCACTCCTTAACACTGACTGGCT-SH] were synthesized by Sangon Biological Engineering Technology and Services Co., Ltd. (Shanghai, China). Ultrapure was purchased from A.S. Watson Group (Hong Kong, China).

### 2.2. Apparatus

The as-prepared BPNSs and PLL-BPNSs composition (PLL-BP) were observed on high resolution transmission electron microscopy (HR-TEM, Tecnai G2 F20, FEI Ltd., Natural Bridge Station, VA, USA) The combination comprising black phosphorus nanosheets and polylysine was characterized using Fourier transform infrared spectroscopy (FTIR, Tensor II, Bruker Ltd., Billerica, MA, USA) All electrochemical experiments were measured on a PGST AT302N Electrochemical Workstation (Metrohm, Herisau, Switzerland). A conventional three electrode system was used, in which a saturated calomel electrode, platinum wire electrode, and bare GCE or Au/PLL-BP/GCE (diameter: 3.0 mm) were adopted as the reference electrode, counter electrode and working electrode, respectively.

### 2.3. Preparation of BPNSs

BPNSs were prepared by exfoliation of BP crystals, as previously reported [35]. Briefly, 5.0 mg of BP crystals were added into 10 mL of ultrapure water solution (containing 1% (*v/v*) Tween-20), and sonicated in an ice bath for 8 h. Then, the obtained brown suspension was centrifuged at 3000× *g* rpm for 30 min, to remove the residual unexfoliated particles, and the supernatant was gathered in the atmosphere of argon. The gathered supernatant was further centrifuged at 8000× *g* rpm for 30 min, and brown-yellow supernatant was collected for future use.

### 2.4. Fabrication of PLL-BPNSs Composition

To synthesize PLL-BPNSs composition, the as-exfoliated BPNSs dispersion was combined with 2 mg/mL PLL (prepared with deoxygenated distilled water), and the resultant mixture solution was agitated for 2 h in a shaker, and then incubated for 12 h at 4 °C.

### 2.5. Preparation of Apt/AuNPs/PLL- BPNSs/GCE Electrode

Bare GCE (diameter: 3 mm) was polished on the polishing cloth with 0.05 μm alumina slurry, and completely cleaned by ultrasonication in ethanol and ultrapure water for 1 min in sequence, followed by drying with argon for later use. The prepared PLL-BPNSs suspension (3 μL) was directly pipetted onto the polished GCE and dried for 12 h at room temperature in an argon atmosphere to obtain PLL-BP/GCE. Then, 3 μL 1 mM AuNPs were dropped onto the PLL-BP complex membrane and dried to form AuNPs/PLL-BP/GCE electrodes under protection of argon. Next, 5 μL of 1 μM activated sulfhydryl primers by TCEP were dropped onto the AuNPs and incubated at room temperature for 40 min. The prepared electrode was placed at room temperature for a period of time, and 5 μL of 1 μM fipronil aptamer was added onto the electrode surface modified with sulfhydryl primer. After the fipronil aptamer reacted with the primer for 1 h, the non-specifically boundfipronil aptamer was gently washed off with PBS buffer (10 mM, pH 7.4) to obtain Apt/primer/AuNPs/PLL- BP/GCE electrodes, as shown in Figure 1.

### 2.6. Electrochemical Detection of Fipronil

The surface of modified aptamer sensor was dripped with 5 μL of fipronil solution at different concentrations (10 pg.mL^−1^–10 μg. mL^−1^), and incubated at room temperature for 60 min. After the fipronil aptamer was fully combined with fipronil in the solution, it was rinsed slightly in phosphate buffer. The modified electrodes were characterized by electrochemical impedance spectroscopy (EIS). EIS was performed within the frequency range between 0.1 to 10^5^ Hz, in 10 mM PBS buffer solutions containing 0.5 mM K_3_[Fe(CN)_6_] and 0.1 M KCl. The concentration of fipronil in the samples was determined by DPV method. The DPV parameters were: scanning range between −0.2 to −0.6 V, amplitude of 0.05 V, pulse width of 0.2 s, and pulse time of 0.5 s.

## 3. Results and Discussion

### 3.1. Characterization of PLL–BPNSs Nanocomposite

The structure of BPNSs prepared in Tween-20 aqueous solution before and after coating with polylysine was analyzed by transmission electron microscopy (TEM).

As shown in Figure 1A, the prepared BPNSs displays free-standing few layer nanosheets with an average sheet diameter of about 600 nm in the low magnification TEM images. Figure 1B–D depicts the fabricated PLL-BPNSs complexes under a low-power transmission electron microscope. BPNSs are scattered in polylysine and well coated with polylysine, and the surface roughness increased significantly when the BPNSs reacted with polylysine. The outer surface of the few-layer BPNSs was completely covered, resulting in morphological alterations.

The combination comprising of black phosphorus nanosheets and polylysine was characterized using Fourier transform infrared spectroscopy (FTIR). As shown in Figure 2, there are two prominent peaks at 1636.96 cm^−1^ and 3450.88 cm^−1^, which correspond to bending and stretching vibrations of NH, respectively. This is a critical indication of successful polylysine formation, and 679.35 cm^−1^ was the out-of-plane absorption peak for CH, suggesting that the black phosphorus nanosheets with polylysine were effectively coupled.

### 3.2. Electrochemical Characterization of Modified Electrodes

It is well known that graphene (GP) and AuNPs are traditional nanomaterials that are widely used in electrochemical sensors, due to their large specific surface area and high conductivity. In order to confirm that black phosphorus nanomaterials have better electron transfer performance compared with other nanomaterials, PLL-BP, PLL-AuNPs, and PLL-GP were obtained by mixing the same concentration of polylysine with the same mass concentration of black phosphorus nanosheet dispersion, gold nanoparticle concentrate, and graphene dispersion, respectively (this mass concentration is calculated from the absorbance of the dispersion). The GCE was modified with these three composite materials, respectively, and completely dried under argon environment. Cyclic voltammetry (CV) was used for characterization in the K_3_[Fe (CN)_6_] mixture system. As shown in Figure 3, the redox peak currents of three composite modified GCE are higher than the bare electrode, but that of PLL-BPNSs/GCE (red line) was much higher than the bare and PLL-Au NPs/GCE, PLL-GP/GCE for K_3_[Fe (CN)_6_] probe, and the peak potential difference (ΔE) was minimum. This means that the PLL-BP modified electrode surface was conducive to the transmission of probe ions on the electrode surface, and the conductivity was greatly enhanced. Thus, it can be shown that, compared with other traditional nanomaterials, black phosphorus nanomaterials have more superior electronic transmission performance, and can be used as an outstanding sensor preparation material in the sensing design strategy for effectively increasing electrical signals.

To investigate the interfacial transmission characteristics of the modified electrode, electrochemical impedance spectroscopy (EIS) tests were performed on GCE, PLL-BP/GCE, AuNPs/PLL-BP/GCE, primer/AuNPs/PLL-BP/GCE, and Apt/primer/AuNPs/PLL-BP/GCE in 10 mM PBS buffer containing 0.1 M KCl and 0.5 mM K_3_ [Fe (CN)_6_]. To match the experimental data, the Randles equivalent circuit model was employed, where Rs denotes the electrolyte resistance, Rct denotes the charge transfer resistance, C denotes the double capacitor, and W is the Warburg impedance. Generally, the semicircle diameter in Nyquist plots for the EIS is equal to the value of electron-transfer resistance (Rct). It can be observed from Figure 4. that the impedance value for the bare GCE decreased after modification by the PLL-BPNSs it decreased, which can be reflected by the reduction in the semicircle diameter (Rct) of the Nyquist curve. Due to the high electron mobility of the BPNSs, the electron transfer rate at the glassy carbon electrode interface increased, and the current hindrance decreased. After further modification with AuNPs, the electron transfer rate was further accelerated due to the huge quantity and large specific surface area of AuNPs, and the impedance semicircle diameter was further reduced. However, once the sulfhydryl primers were covalently modified to the electrode surface through Au-S bonds, the DNA primers were unable to transmit electricity, and the addition of DNA primers increased steric hindrance. This hindered the electron transfer, thus the measured impedance value for the electrode surface increased significantly. Finally, when the DNA primers were hybridized with free fipronil aptamers in the solution, the semicircle diameter of the Nyquist curve further increased, and the measured impedance was at maximum. This was because a specific combination of aptamers and primers further increased the steric hindrance of the electrode, and the sensor was successfully established and verified.

### 3.3. Optimization of Recognition Conditions

AuNPs and thiol primers interact through the formation stably Au-S covalent bond. In this study, AuNPs were modified on the electrode coated with PLL-BPNSs composite to further amplify the electrical signal. The large specific surface area of AuNPs makes for stable binding with thiol primers. Different doses of AuNPs, 100, 200, 300, 400, and 500 ng, were, respectively, added to the prepared PLL-BP/GCE electrode surface and dried, and DPV measurements were performed with electrodes in the K_3_ [Fe (CN)_6_] mixed system. The results are shown in Figure 5A. As can be seen, the current increases with increasing amounts of AuNPs, reaching a maximum at 300 ng, followed by a montonic decrease as the amount of gold is increased. This is an indication that at more than 300 ng, the AuNPs form a thick layer, which results in a decrease in the peak current.

Thethiol primers play the role of bridge between aptamer and electrode surface, so the incubation time for thiol primers on the electrode also determines the performance of the aptamer sensor. A total of 5 μL of 1 μM thiol primers activated by 1 mM TCEP were added onto AuNPs films on different AuNPs/PLL-BP/GCE electrodes, and then incubated on the electrode for 10–70 min, respectively. Similarly, DPV detection was performed in the K_3_[Fe (CN)_6_] mixture system. The ΔI_peak_ for DPV shown in Figure 5B is the difference between the actual measured value and the blank background current, which is an indication of the binding of the primers on the electrode surface. As can be seen in Figure 5B, the primer binds to the electrode surface in about 50 min, as shown by the leveling of the current value.

The aptamers and primers were effectively combined by hydrogen bonding, and the aptamers were enriched on the electrode surface via hydrogen bonding between base pairs. In order to reduce the preparation time for the sensor, the bonding time between fipronil aptamer and mercapto primer should also be optimized. The prepared Primer/AuNPs/PLL-BP/GCE electrode surface was blocked with MCH blocking solution for half an hour. The blocking solution blocked the remaining active sites on the electrode surface to prevent its non-specific adsorption. A total of 5 μL of 1 μM fipronil aptamers were dropped onto the modified electrode surface and incubated for 10–70 min, respectively, and the incubation progress was detected. As shown in Figure 5C, the peak change value for the DPV current steadily reduced with time in the K_3_[Fe (CN)_6_] mixed solution system, and stabilized after 50 min of incubation. It can be seen that, the primers on the electrode surface were sufficiently combined with the fipronil aptamer, and the bonding of the aptamer led to greater biological molecular weight on the electrode surface, which made the electrolysis system gradually stable. Therefore, it was concluded that the optimal incubation time for fipronil aptamer was 50 min.

### 3.4. Detection of Fipronil on Apt/Primer/AuNPs/PLL-BP/GCE

When the electrolyte solution contains fipronil pesticide, the fipronil aptamer can specifically recognize and capture fipronil in the solution, which makes the structure and morphology of the fipronil aptamer change. The binding of fipronil to aptamer generates the spatial conformation of aptamer. Due to the effect of spatial conformation, the aptamer is separated from the primer and results in the change of the surface state of the electrode. As a result, the electronic transmission obstacle decreases, and the peak current for the detection system increases. Figure 6A shows the current response of Apt/primer/AuNPs/PLL-BP/GCE to different concentrations of fipronil. The current gradually increased with increased fipronil concentration. The reason is that more aptamers were specifically bound to fipronil and separated from the primers with increased fipronil concentration, which accelerated the electron transfer on the sensor surface. As shown in Figure 6B, the logarithmic values for ∆I_DPV_ and C_Fp_ for the fipronil aptamer sensor were positively correlated across concentration gradient of 0.1 ng/mL–10 μg/mL fipronil, and the sensor’s linear fitting equation was: ∆I (μA) = 13.04 logC + 22.35, with a linear correlation coefficient R^2^ of 0.998, and detection limit at 74 pg/mL (0.17 nM).

### 3.5. Stability, Reproducibility and Specificity

Under the parameters described above, fipronil concentrations of 1 ng/mL were measured six times in parallel, and the peak current for the detection resulted in a relative standard deviation of about 2.39%, with repeatability meeting the criteria of usage. To evaluate the stability of the aptasensor, the PLL-BP-Apt sensor was used to measure the current response at various time intervals, namely by adding 10 μg mL^−1^ fipronil to PBS samples on day 0, day 7, and day 14. These findings indicated that the current of fipronil in PBS did not substantially drop after 14 days (26.43%), suggesting that the developed aptasensor exhibited high storage stability.

The sensor’s specificity is also critical. If the aptamer sensor matches, the detector’s specificity for fipronil must be confirmed using other pesticide interferents. The blank was the sensor without fipronil in PBS buffer, and 10 ng/mL aldicarb sulfoxide (As), fenitrothion (Ft), carbofuran (Cf), aldicarb (Ac), and fipronil (Fp) were employed in the test. For the specificity study, the aptamer sensor was incubated with the four pesticides listed above and a mixture (Mix) comprising the target assay (Fp) and the mixture. The specificity was assessed by the difference between the peak DPV of each pesticide compared to the blank. As seen in Figure 7, the experimental group containing fipronil solution and a combination of pesticides, including fipronil, exhibited much greater fluctuations in the DPV peak than the other pesticides, showing that the sensor has a greater selectivity for fipronil.

### 3.6. Sample Analysis and Analyte Detection

To validate the sensor’s effectiveness in actual sample detection, we used the standard addition method to detect fipronil samples at the following concentrations; 10 ng/mL, 100 ng/mL, and 1 μg/mL in Shenlong Lake water samples. The recoveries were 107.9%, 100.69%, and 84.1%, respectively, with relative standard deviation (RSD) (n = 3) of 0.61%, 8.23%, and 5.58%, as shown in Table 1. The two-dimensional sensing platform created provides a high degree of selectivity and sensitivity. Simultaneously, we also compared the experimental results with those from other methods described in recent reports. As shown in Table 2, this method demonstrated a relatively wide linear range and a comparable limit of detection with the other methods. Although the sensor in the literature [36] has a low detection limit for fipronil, it uses a high-cost small molecule antigen antibody and is difficult to prepare. Another sensor with a lower detection limit in the literature [37] uses molecular imprinting, which can detect only a narrow range of concentrations. Thus, the developed method is well applicable for fipronil detection in real samples.

## 4. Conclusions

In summary, a new aptasensor based on PLL-BPNSs and AuNPs is fabricated. PLL-BPNSs was prepared using PLL, which functionalized the surface of BP nanosheets by non-covalent interaction. AuNPs bonded to PLL-BPNSs firmly to give AuNPs/PLL-BP by assembling. The assembled AuNPs/PLL-BP matrix is desirable for increasing electron transfer and bonding thio primers. The particular DNA aptamers were used targeting and binding fipronil. The chosen DNA aptamers were effectively combined with primers and enriched on the electrode surface via hydrogen bonding between base pairs. The AuNPs/PLL-BP nanostructure DNA aptasensor is a label-free electrochemical sensing platform for fipronil. The electron transport was detected on the primer/AuNPs/PLL-BP electrode in K_3_[Fe (CN)_6_] mixed liquid system. Through the synergistic impact of chosen high affinity aptamers and increased electrochemical performance of nanostructures, the sensor displayed great selectivity and sensitivity. The detection limit of the sensor was 74 pg/mL, and the linear range was 0.1 ng/mL–10 μg/mL. Our discovery brings up numerous fascinating options to develop improved rapid detection of fipronil.

## Data Availability

Not applicable.

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
