# Peer review of "Label-Free Aptasensor for Detection of Fipronil Based on Black Phosphorus Nanosheets"

_biosensors, 2022, doi:10.3390/bios12100775_

Round 1
Reviewer 1 Report
In this study, Huang et al., reported an aptamer-conjugated black phosphorus nanosheet (BP-NS) sensor for the detection of fipronil, a phenylpyrazole pesticide. While the authors have overall demonstrated the function of this electrochemical sensor, there are still several clear major and minor concerns.
Major concerns:
1. The authors should do a much better job in summarizing existing fipronil sensors. A long list of aptamer-based fipronil sensors have been reported previously, especially fluorescence-based ones. The authors should carefully compare this BP-NS sensor with those existing ones.
2. In several places, the explanations of the mechanism of signal responses don’t seem to make sense. For example, fipronil-induced electronic transmission signal change was explained as “due to the effect of steric hindrance, the aptamer is separated from the primer”, which doesn’t seem to be true. Based on the sequence of aptamer and primer, there are 20 base pairs in between, which is quite stable under the standard experimental condition. Any evidence, like gel electrophoresis or collection of released aptamers can demonstrate the real separation between these two strands. Similarly, the authors indicated that “the chosen DNA aptamer was attached to PLL-BP by Coulomb contact”, while it seems that Au-thiol interactions should be the more dominant force here.
Minor issues:
3. Fig. 1 TEM is confusing. In Fig. 1B, it will be better to show the region including the interface between BPNS and surface, as shown in Reference 20. What are the current dark background shown?
4. The authors tried to compare the signal from BPNS, AuNP, and graphene, what does the “same amount” of each dispersion mean? The same molar concentration, the same mass? Of starting materials or final nanoparticles? Without this information, it is hard to conclude that BPNS “have more superior electronic transmission performance” just based on Fig. 3.
3. Throughout the manuscript, the authors should indicate how many repeated experiments were performed for each figure shown.
6. The meaning of error bars should also be explained, especially in Fig. 5, 6B,
7. References were missing in several places, such as “It is well-known that thiol primers play the role of …”, “the detection limit for the aptasensor based on the PLL-BP nanostructure was lower than …”
8. The meaning of “RSD” relative standard deviation should be explained.
9. Table 1 was duplicated.
10. The authors should also pay more attention to the significant figures of their results, such as 74.16 pg/mL likely should be 74 pg/mL.
Reviewer 2 Report
This paper entitled: Label-free aptasensor for detection of fipronil based on black phosphorus nanosheets describes a sensor based on the functionalization of a modified surface of a glassy carbon electrode with gold nanoparticles, thiolate primers and fipronil aptamer for the electrochemical detection of fipronil in river water. This approach is novel and would be a useful addition to the other methods that have been described for the detection of fipronil in environmental samples. However, there are several deficiencies in the current version of this paper which must be addressed before it is considered for publication.
Main Comments
The abstract should be re-written to reflect the main findings from the investigations. Based on the evidence presented, there is no basis for claiming that the developed sensor offers outstanding repeatability, stability, and specificity. Quote the actual figures of merit of the technique found and let the readers judge the performance of the sensor.
The description of the optimization of the measurement conditions should be re-written. Particularly lines 234-245 in which the processes occurring on the electrode surface are discussed. The description should be broken-up by using the different parts of the curves in Figure 5.
The conclusion does not reflect the findings from the investigations. It should be re-written to point at the salient features of the sensor and the authors should speculate on potential developments.
Corrections
Line 49 should read: …. and can be prepared
Line 109-111 rewrite the sentence so that it is clear what you are trying to say.
Line 128-129 should read: …. The non-specifically bound fipronil aptamer was gently washed off.
Line 158 re-write the legend to Figure 1 and point out the salient features.
Line160-161 should read: …. there are two prominent peaks at 1689.96 and 3540.88 cm-1, ….and stretching vibrations of NH, respectively.
Line 175 What does CV stand for?
Line 185 Add detail to the Fig 3 legend so that you can direct the reader to the key features.
Line 207 should read: …. the measured impedance is at maximum.
Line 215 should read: The AuNPs and thiol primers interact through the formation of stable Au-S covalent bonds.
Line217-219 should read: The large specific surface area of the AuNPs make for stable binding with the thiol primers.
Line 219 should read: Different doses of AuNPs, 100, 200, 300, 400 and 500ng, respectively were added to the prepared PLL-BP/GCE electrode surface, dried, and DPV measurements were performed with the electrodes in the K3[Fe(CN)6] mixed system. The results are shown in Fig5A. As can be seen the current increase with increasing amounts of AuNPs reaching a maximum at 300ng followed by a monotonic decrease as the amount of gold is increased. An indication that at more than 300ng, the AuNPs form a thick layer which results in decrease of the peak current.
Line 235-236 should read: …. Which is an indication of the binding of the primers to the electrode surface.
Line 236-237 As can be seen in Figs 5B and 5C the primer binds to the electrode surface in about 50mins as shown by the levelling of the current value.
Line 256-258 There is no evidence to support the claim that the primers are completely bound to the fipronil aptamer.
Line 258 How was the increase in the biological molecular weight established?
Line 312 Replace specificity with selectivity.
Table 1 delete the bottom part of the table as the information is replicated.
Round 2
Reviewer 1 Report
The authors did a fine job in revision. Some of my original comments were addressed. However, concerns still exist: in my comment #4, what does "same concentration” really mean is still convincing, Comments #5 and #6 were not carefully addressed.
Author Response
Honored teacher:
Thank you for your comments on the revision of our manuscript, In response to your questions, we have given the following answers,perhaps we did not understand your questions well. If there are still mistakes, please do not hesitate to contact us.
Question:4.The authors tried to compare the signal from BPNS, AuNP, and graphene, what does the “same amount” of each dispersion mean? The same molar concentration, the same mass? Of starting materials or final nanoparticles? Without this information, it is hard to conclude that BPNS “have more superior electronic transmission performance” just based on Fig. 3.
Answer:It is “same mass concentration”, (This mass concentration is calculated from the absorbance of the dispersion).
Question:5.Throughout the manuscript, the authors should indicate how many repeated experiments were performed for each figure shown.
Answer:Thank you for your suggestions.We explain this in the title of Figure 5, 6,that is“Every point in the graph represents the mean of three successive measurements (n = 3) at each weight or time. ”
Question:6. The meaning of error bars should also be explained, especially in Fig. 5, 6B.
Answer: We explain this in the title of Figure 5, 6,"The length of the error bar, also known as the size of the standard deviation, describes how clustered data points are around the mean"
